# Physiological and Productivity Responses in Two Chili Pepper Morphotypes (*Capsicum annuum* L.) under Different Soil Moisture Contents

Aurelio Pedroza-Sandoval [1,*], José Rafael Minjares-Fuentes [2] , Ricardo Trejo-Calzada [1] and Isaac Gramillo-Avila [1]

1    Unidad Regional Universitaria de Zonas Áridas, Universidad Autónoma Chapingo, Km. 40 Carr. Gómez Palacio—Chihuahua, Bermejillo C.P. 35230, Durango, Mexico; rtrejo@chapingo.uruza.edu.mx (R.T.-C.); igramillo4@gmail.com (I.G.-A.)
2    Facultad de Ciencias Químicas, Universidad Juárez, Estado de Durango, Av. Artículo 123 s/n. Fracc. Filadelfia, Gómez Palacio C.P. 35010, Durango, Mexico; rafael.minjares@ujed.mx
*    Correspondence: apedroza@chapingo.uruza.edu.mx

**Abstract:** The aim of this study was to explore some physiological and productivity responses of two chili pepper morphotypes (*Capsicum annuum* L.) exposed to different soil moisture contents. A randomized block design in a split-plot arrangement with four replicates was used. The large plots (32 m long and 3.2 m width) were 25% ± 2 as the optimum soil moisture content (OSMC), and 20% ± 2 as the suboptimum soil moisture content (SSMC); the small plots (16 m long and 3.2 m width) were two chili pepper morphotypes: Jalapeño and Chilaca, respectively. Jalapeño plants showed more stability in relative water content (RWC), photosynthetic activity ($\mu$mol $CO_2$/m$^2$/s), and a relatively low transpiration (mmol $H_2O$/m$^2$/s) and stomatal conductance ($\mu$mol $H_2O$/m$^2$/s); therefore, it had a higher number of flowers per plant and number of fruits per plant, consequently recording a high fruit production of 3.94 and 2.99 kg/m$^2$ in OSMC and SSMC, respectively. In contrast, the Chilaca chili showed low stability in water relative content (WRC), photosynthesis, and transpiration, going from OSMC to SSMC, as well as showed a lower yield in SSMC; however, all of that was compensated by its size and weight of the fruit per plant, with a yield of 4.95 kg/m$^2$ in OSMC. Therefore, the Jalapeño chili pepper could be an option when the irrigation water is limited, and the Chilaca chili pepper when this resource is not limited.

**Keywords:** drought; water scarcity; plant physiology; agri-food; arid lands

## 1. Introduction

Fruits and vegetables are an essential part of the diet of the human population due to their nutritional contribution. In 2021, the United Nations declared it as the international year of fruits and vegetables, spotlighting their basic role in human nutrition and food security, as well as the importance to improve the sustainable production of these crops [1]. These plant species are the most common for the healthy nutrition and include a varied group of plant foods with important energy and nutrient and fiber content [2,3]. Furthermore, these agri-food crops are a source of phytochemicals with antioxidant activity, and the processes for obtaining these products, as well as its effectiveness, mechanisms, and application methods, are still being studied [4,5] The intake of these food components are related to a lower incidence of human diseases to improve the health and other benefits [6–8].

Chili peppers, along with beans and maize, are the oldest crops cultivated in the Americas [9]. Chili peppers are vegetable species that grow in different parts of the world and are used as a dietary supplement. These kinds of plants are native to the Americas and are cultivated in warm climates around the world, with a great genetic diversity in

Mexico [10]. Chili peppers can be eaten fresh or dried and are used to make chili powder to flavor barbecue, hot curry, and other spicy sauces [11], but they are also used as a pharmaceutical product for health [12]. There are different species of chili pepper, but *Capsicum annuum* L. is the species of greatest commercial importance in the world, with a production of 24 million tons per year [13].

In Mexico, the chili pepper crop has a social and economic importance due to its place in the nation's gastronomy and its high demand as an agri-food in the market. *Capsicum annuum* is produced practically in all the states of Mexico, covering an area of 129,325.4 ha, with an average yield of 21.8 tons/ha and a production of 122.49 thousand tons per year in the country [14].

According to Sanchez Toledano et al. [15], the Jalapeño pepper is one of the best-known and mostly consumed chili peppers in Mexico, but other pepper species are demanded in the national market, considering different decision criteria such as the physical appearance as well as flavor, color, and size, which are highly valued by consumers. The same authors cite that, although there are similar consumption patterns, purchasing behaviors change by region, which is important to set up strategies in the production and marketing.

Chili pepper growth and productivity is influenced by the environment and crop management. It is sensitive to low temperatures with an optimum of 20–26 °C, and it requires fertile soils with mulch, adequate plant nutrition, and permanent water supply due to its high sensitivity to water deficits [16]. Water availability is one of the biggest risks in crop production, due to the high frequency of droughts in the main hydrological watershed of agricultural areas [17].

Droughts and the overexploitation of aquifer water have been prevalent for decades in the different irrigated agricultural areas in Mexico, with consequences not only in the amount of available water, but also in the water quality. Scarcity and chemical contamination of the water are part of the environmental impact on the agroecosystem, with a negative effect on productivity and a high risk to health via consumption of contaminated agrifood products [18].

Different producer regions of chili pepper in northern Mexico are being adversely impacted by high water consumption to maintain adequate levels of productivity [19]. These production systems make intensive use of natural resources such as water and soil. Some crop management alternatives are needed to produce chili peppers with lower amounts of water. Techniques such as biostimulants are used during critical phenological stages of the crop [20,21], as well as soil moisture retainers [22] and plant species tolerant to water deficits [23,24], all of which are proving to be options to mitigate water shortage, mainly with the evaluation of chili pepper cultivars and its tolerance to water stress in terms of physiology, biochemical, growth, and yield responses [25–27].

Water holding capacity (WHC) in the soil is an important agronomic indicator that play a key role in plant growth and development. WHC depends on soil texture type [28]. Different irrigation procedures are being tested to evaluate water deficit tolerance in plants. Thus, McCoy et al. [29] reported the response to water deficits and the adaptation to drought of a chili pepper germplasm. Macias-Bobadilla et al. [30] evaluated four treatments of irrigation: gradual water deficit (GWD), initial waterlogging with gradual water deficit (IWGD), sudden water deficit with gradual recovery (SWDR), and no deficit of water (NDW) on morphological, physiological, and genetic responses of chili pepper. Mahmood et al. [31] evaluated two levels of drought at a 35% and 65% field capacity (FC) to identify the response of capsaicin content and antioxidant activity in different genotypes of chili pepper. Agyemang et al. [32] investigated the effect of different water supply treatments: rain-fed (RF), deficit irrigation (DI), and optimum water supply (OWS) on the physiological response and phytonutrients in chili pepper cultivars. Quintana et al. [33] exposed habanero pepper (*Capsicum chinense* Jacq.) to five levels of substrate water availability (60, 50, 40, 30, and 20%), assessing the leaf water potential, plant growth, biomass distribution, fruit yield, fruit size, and harvest index; meanwhile, Yildirim et al. [34] applied three different irrigation levels of 100%, 75%, and 50% of the water to reach the FC and its effect in the

plant growth, stoma conductivity, relative water content, and antioxidant contents as a response to water stress in chili pepper cultivars.

In northern Mexico, different types of *C. annuum* L. are grown; among them, bell pepper, Chilaca, and Jalapeño chili peppers stand out. Each one requires different management practices and water consumption levels [35]. Identifying the best chili pepper type to grow in each region based on agroclimatic potential is important to enhance the regional productivity [36]. For the economic importance in the region, and the necessity to perform a more efficient use of water, the Chilaca and Jalapeño peppers were evaluated in this study. Chilaca pepper fruit have an intense and shiny green color that changes to dark brown at maturity; it measures between 15 and 30 cm long and 2–3 cm wide; and it is medium spicy. The plants can reach 60 cm high and are produced mainly in the north-central area of Mexico [37]. Meanwhile, Jalapeño pepper fruit is green and changes to an intense red color at maturity; it has an elongated conical shape, measuring on average 6 cm long by 2.5 cm wide; and when processed, it becomes a chipotle pepper. The plant measures from 80 cm to 1.50 m. Jalapeño chili is the most widely grown pepper chili crop in Mexico and more than 60% of the production is processed into sauces, sausage chilies, and dried chilies [38]. Thus, the aim of this study was to evaluate the Chilaca and Jalapeño chili pepper morphotypes (*Capsicum annuum* L.) in some physiological and productive indicators as a response to different watered regimes in northern Mexico.

## 2. Materials and Methods

### 2.1. Geographical Location

The study was carried out in 2021 in the experimental area of the Unidad Regional Universitaria de Zonas Áridas (URUZA), Universidad Autónoma Chapingo (UACh), in Bermejillo, Mapimí, Durango, Mexico. The region is located at 24° 22′ and 26° 23′ NL and 101° 41′ and 104° 61′ WL at an elevation of 1100 m (Figure 1). The climate is dry, with rainfall in summer and cool winters, an average annual rainfall of 250 mm, and an annual temperature of 21 °C [39].

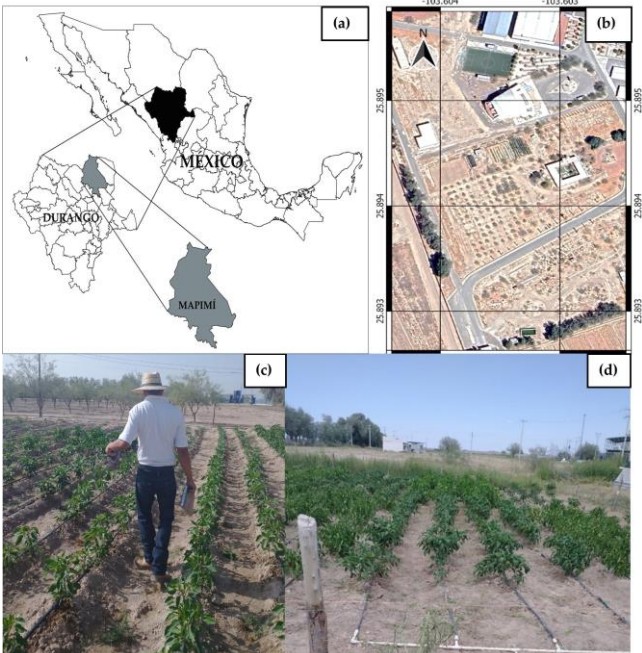

**Figure 1.** Geographic location of the study area in the Mapimí, Durango, Mexico: (**a**) Map of Mexico, Durango Sate, and Mapimi Muncipaliy; (**b**) geographical location of the experimental area; and (**c**,**d**) different views of experimental area.

## 2.2. Experimental Design and Treatments

A randomized block design in a split-plot arrangement with four replicates was used. The large plots (16 m long and 3.2 m width) were of an optimum soil moisture content (OSMC) corresponding to 25% ± 2, and a suboptimum soil moisture content (SSMC) corresponding to 20% ± 2. Both large plots were located in a randomized way in parallel for each replication, which makes it easier to apply irrigation at specific times for each moisture content. This is the most common experimental design used in these types of studies [30], with small plots (8 m long and 3.2 m wide) within each large plot were the Jalapeño and Chilaca chili pepper morphotypes, respectively. Four treatments were obtained via factorial 2 × 2, where the first-one factor was the soil moisture content: OSMC and SSMC, and the second-one factor was the chili pepper morphotypes: Jalapeño and Chilaca. The experimental unit was integrated via 4 rows, 8 m long and 0.8 m wide apart, while the useful plots (UP) were the two middle rows (2 × 0.8 = 1.6 m wide and 8 m long per UP). Four plants were randomly selected in each UP for analysis.

Before establishing the study in the experimental area on 1 April 2021, the sowing was carried out with certified seed of each chili pepper morphotype in germination trays using peat moss as a substrate under shade mesh conditions. Forty days after sowing, the germinated seedlings (25 cm height) were transplanted into the experimental field to 30 cm between plants. This resulted in rows 0.8 m wide and 8 m long, with four rows by treatment, and, accordingly, the plant density was 4.16 plant/m$^2$.

According to the soil analysis of the experimental area, it corresponds to a sandy loam soil with 56% sand, 6% clay, and 36% silt, with a pH of 8.08, classifying it as moderately alkaline; an electrical conductivity of 5.6 dS/m, classifying it as saline soil; a cation exchange capacity of 13.69 meq/100 g; and a sodium absorption ratio (SAR) and exchangeable sodium percentage (ESP) of 8.56/11.21, respectively.

## 2.3. Soil Moisture Content

According to the soil water retention curve calculated by the membrane pot method [40], the field capacity (FC) was 26.1% and the permanent wilting point (PWP) was 13.1% (Figure 2), which is equivalent to 0.261 m$^3$/m$^3$ (−0.03 MPa) and 0.131 m$^3$/m$^3$ (−1.5 MPa), respectively [41]. This behavior of the soil water retention curve is common in sandy loam soils.

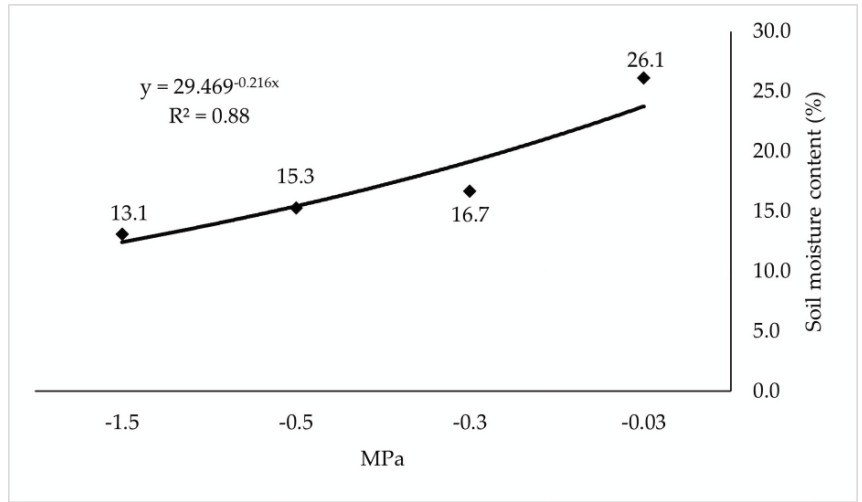

**Figure 2.** Soil water retention curve determined by the membrane pot method [40].

Therefore, the optimum soil moisture content (OSMC) had a range from 23 to 27% (25% ± 2), and the suboptimum soil moisture content (SSMC) from 18 to 22% (20% ± 2). According to the soil's physical parameters of FC (26.1%) and PWP (13.1%), the OSMC is 100% FC and the SSMC is 81.5% FC, with both corresponding to the upper limits of each

range by 27 and 22%, respectively. These soil contents, according to those reported by Mahmood et al. [31] and Quintana et al. [33], who evaluated 35–65% FC and 50, 75, and 100% FC, respectively, reported severe damage to the growth and development of the chili plant at 65% FC or less. The field capacity cannot be lowered too much, since the chili plant has a $C_3$ photosynthetic pathway that makes it sensitive to water deficits [29].

### 2.4. Experimental Setup

Drip irrigation system was used. This irrigation system ensures that the plants will be nurtured and watered on a regular basis with appropriate amounts of hydration and also allows us to set up a nutrient delivery system [42]. It is one of the most efficient systems for water use, which is important in arid lands. The irrigation treatments were applied via the main irrigation line (2″) with perpendicular irrigation lines (1/2″) per row. The irrigation supplies were controlled using on–off valves. One self-compensating dripper (Model: CHAPIN DRIP TAPE, JAIN Irrigation Inc., Watertown, NY, USA) that emitted 2 L/h was used on each plant. At the start of the experiment, the irrigation treatments were watered to FC. The soil moisture content was determined regularly in real-time with a digital tensiometer (Model: MO750, Extech Instruments Co., Laredo, TX, USA). When the soil moisture content reached the lowest limit of a treatment, irrigation was resumed until the upper limit of each irrigation treatment. The recovery irrigation took around 4 h for each treatment and every four days between irrigation times, approximately (Figure 3).

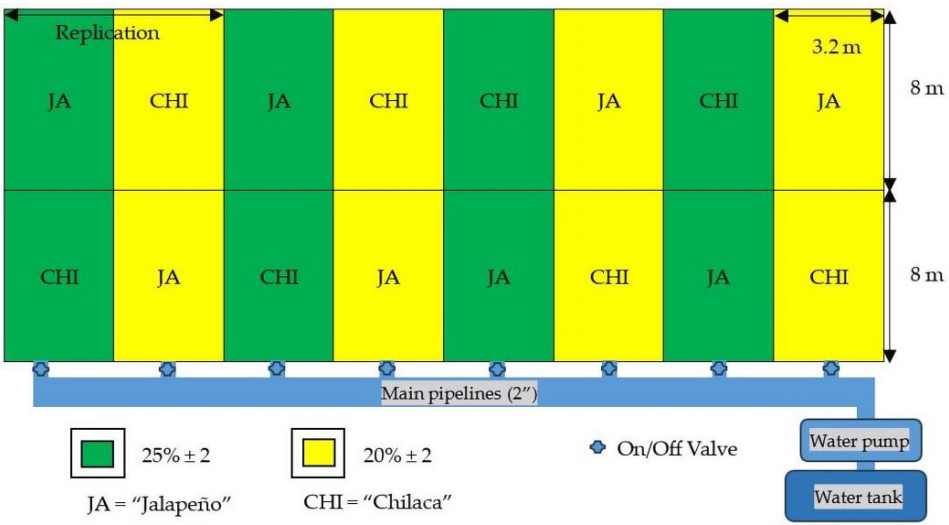

**Figure 3.** Experimental area of evaluation of two chili pepper morphotypes of *C. annuum* L. under optimum soil moisture content (25 ± 2%) and suboptimum soil moisture content (20 ± 2%).

### 2.5. Measured Variables

2.5.1. Climate and Soil Variables

Rainfall (mm) and temperature (°C) in Bermejillo, Durango, México were recorded, using a microclimatic station (Model: 6162, Davis Instruments, Hayward, CA, USA) during 2021.

2.5.2. Physiological Variables

Relative water content (%) (RWC). This variable was measured three times (21 June, 30 August, and 1 October 2021) corresponding to vegetative and flowering and fruiting plant stages, respectively, between 10:00 and 11:00 a.m. on each evaluation date, taking a complete leaf from the fourth node from top to bottom of the plant, and the calculation was according to the following equation [43]:

$$RWC = \left[\frac{\text{Fresh weight} - \text{Dry weight}}{\text{Saturated weight} - \text{Dry weight}}\right] * [100] \tag{1}$$

Photosynthesis ($\mu$mol $CO_2$ /$m^2$/s), transpiration (mmol $H_2O$/ $m^2$/s), and stomatal conductance ($\mu$mol $H_2O$/$m^2$/s) were recorded by using a portable photosynthesis device with infrared ray gas analyzers from Brand LI-6400 (LI-COR®, Inc. Lincoln, NE, USA). These variables were measured on 10 June and 10 July 2021, during the flowering and fruiting stages, respectively. These evaluation dates were chosen since they correspond to the phenological stages most sensitive to environmental stress in most of the plants [44].

### 2.5.3. Productive Variables

The accumulated number of flower buds per plant, accumulated flowers per plant, and accumulated number of fruits per plant, as well as accumulated weight of fruits of chili pepper per plant (g), were assessed. These variables corresponded to the accumulation of 10 evaluation dates with 10-day intervals between them. Finally, pepper yield/$m^2$ was calculated using the following formula:

$$PYSM = [WHFP] * [4.16] \tag{2}$$

where PYSM is the pepper yield per square meter (kg), WHFP, is the weight of the harvested fruits per plant, and 4.16 is the plant density/$m^2$.

### 2.6. Statistical Analysis

The database was analyzed with the GLM procedure of the statistical analysis system and Tukey's test ($p \leq 0.05$) using SAS Institute v. 9.0, software (Cary, NC, USA), and regression analysis was made using Excell V. 7.0 Program.

## 3. Results

### 3.1. Climate Conditions

In 2021, the annual accumulated pluvial precipitation was 109.2 mm. This value is very low compared to the historical average of 250 mm, being a particularly dry year. During the experiment from April to October, there was a total rainfall of 105.4 mm, but July registered the higher rainfall with 54.4 mm. Meanwhile, the monthly mean temperature varied from 27.6 to 25.9 °C from June to August, respectively, with a maximum mean temperature of 27.6 °C in June (Figure 4).

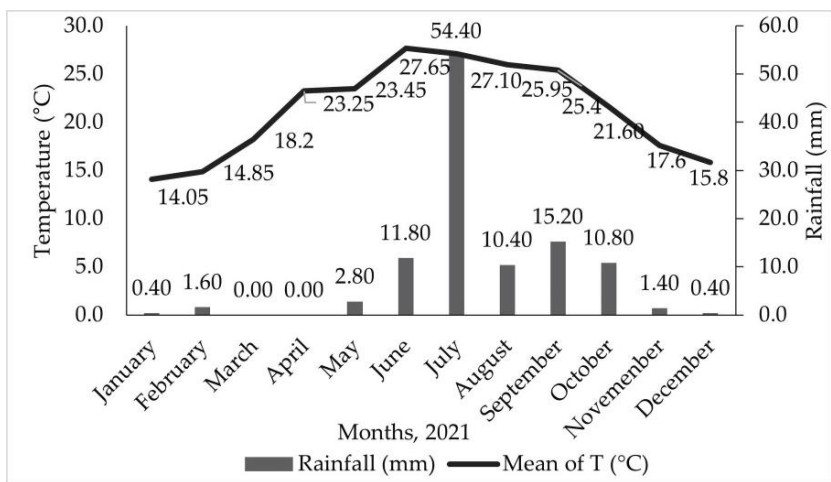

**Figure 4.** Temperature (°C) and Rainfall (mm) in Bermejillo, Mapimí, Dgo. Mexico, in 2021. Source: Experimental Station of the Unidad Regional Universitaria de Zonas Áridas, UACh.

### 3.2. Physiological Variables

### 3.2.1. Relative Water Content (RWC)

Both chili pepper morphotypes showed a similar RWC (~60%) when the plants were grown under OSMC (25% $\pm$ 2), which is a value relatively low but enough to maintain the

growth in both morphotypes' chili peppers [45]. RWC in Chilaca was lower in SSMC with a difference of 2.7% regarding Jalapeño (Figure 5).

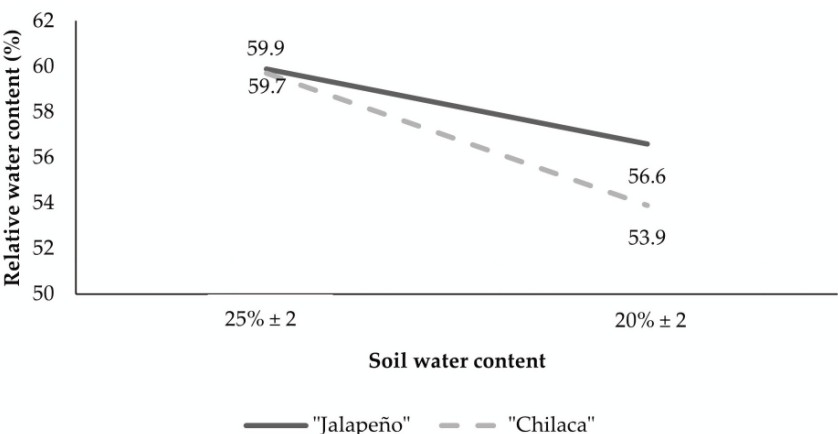

**Figure 5.** Relative water content (RWC) of two chili pepper morphotypes grown under different watered regimes.

### 3.2.2. Photosynthesis

The photosynthesis rate was statistically different ($p \leq 0.05$) between the chili morphotypes at each moisture content within each evaluation date. A greater photosynthetic efficiency was observed at 30 DAT (flower buds' stage), compared to 75 DAT (flowering stage and onset of fruiting), with average values of 17.6 and 14.6 µmol $CO_2/m^2/s$, respectively. The Chilaca chili pepper had the best response (21.8 µmol $CO_2/m^2/s$) in OSMC at 30 DAT, while the Jalapeño pepper was in SSMC (22 µmol $CO_2/m^2/s$) at 75 DAT. The lowest photosynthesis rate was recorded in SSMC in the Chilaca chili pepper at 75 DAT with a value of 4.5 µmol $CO_2/m^2/s$, while the Jalapeño chili pepper showed 13.9 µmol $CO_2/m^2/s$. Thus, Jalapeño had 308.8% more photosynthesis under SSMC as compared to Chilaca (Figure 6).

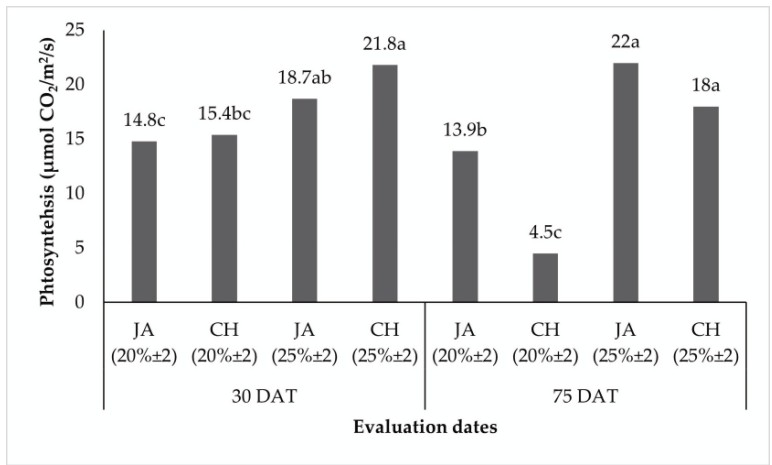

**Figure 6.** Photosynthesis behavior of two chili pepper morphotypes (*Capsicum annuum* L.) in different soil moisture contents on two evaluation dates. Tukey test ($p \leq 0.05$). Numbers with the same letter on the bars in each evaluation date are statistically equal. CH = Chilaca; JA = Jalapeño; DAT = Days after transplanting.

### 3.2.3. Transpiration

Transpiration (mmol $H_2O$ m$^2$/s) behaved similarly to the photosynthesis. Transpiration was higher in OSMC and lower in SSMC in both evaluation dates, but also with a drastic reduction in the Chilaca chili in SSMC with values from 4.7 to 1.8 mmol $H_2O/m^2/s$

at 30 and 75 DAT, respectively, compared to the Jalapeño chili with values from 4.7 to 4.4 mmol $H_2O/m^2/s$ to 30 and 75 DAT, respectively. In the same sense, the Chilaca chili pepper had the best response in OSMC at 30 DAT and the Jalapeño chili pepper in SSMC at 75 DAT, with values of 7.7 and 8.5 mmol $H_2O/m^2/s$, respectively. The Chilaca chili pepper showed a lower transpiration rate under SSMC at 75 DDT since Jalapeño had a transpiration rate that was 244.4% greater under the same conditions (Figure 7).

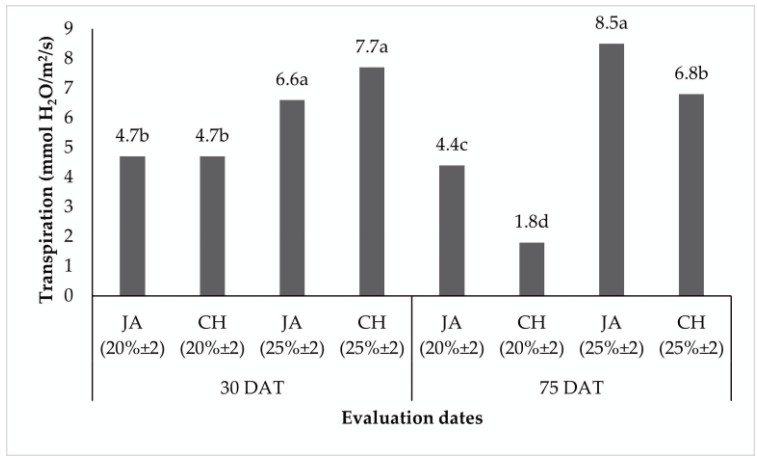

**Figure 7.** Transpiration behavior of two chili pepper phototypes (*Capsicum annuum* L.) in different soil moisture contents on two evaluation dates. Tukey test ($p \leq 0.05$). Numbers with the same letter on the bars in each evaluation date are statistically equal. CH = Chilaca; JA = Jalapeño; DAT = Days after transplanting.

### 3.2.4. Stomata Conductance

Stomatal conductance behavior was very similar to that of photosynthesis and transpiration, with a drastic decrease in stomatal closure in the Chilaca chili, with stomatal conductance values of 0.139 to 0.05 μmol $H_2O\ m^2/s$ at 30 and 75 DAT, respectively, while the Jalapeño pepper remained more stable with values of 0.136 to 0.137 μmol $H_2O\ m^2/s$, respectively, with both under SSMC. The best response was registered in the Jalapeño chili pepper with a value of 0.313 μmol $H_2O/m^2/s$, and the worst response was in the Chilaca chili pepper in SSMC at 75 DAT (Figure 8).

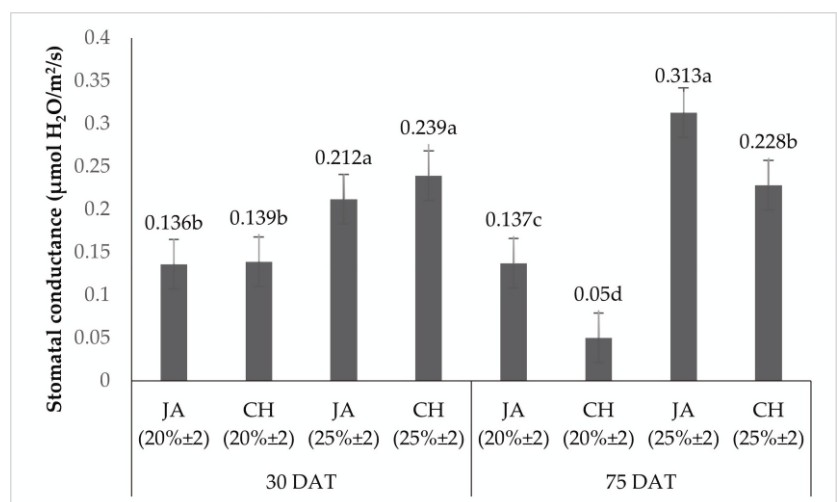

**Figure 8.** Stomatal conductance behavior of two chili pepper phototypes (*Capsicum annuum* L.) in different soil moisture contents on two evaluation dates. Tukey test ($p \leq 0.05$). Numbers with the same letter on the bars in each evaluation date are statistically equal. CH = Chilaca; JA = Jalapeño; DAT = Days after transplanting.

*3.3. Productivity and Yield Variables*

The accumulated number of flower buds did not vary statistically ($p \leq 0.05$) between chili morphotypes and neither in the soil moisture contents at 30 DAT with an average value of 173.25. However, the accumulated number of flowers fell drastically at 75 DAT to an average of 54.45, corresponding to a 68.57% lower number of flowers than of the number of flower bud. Chilaca was more affected in the accumulated flowers per plant with values of 30.7 and 62.8 in SSMC and OSMC, respectively, while Jalapeño was more stable with values of 54 and 70.3 in SSMC and OSMC, respectively (Figure 9).

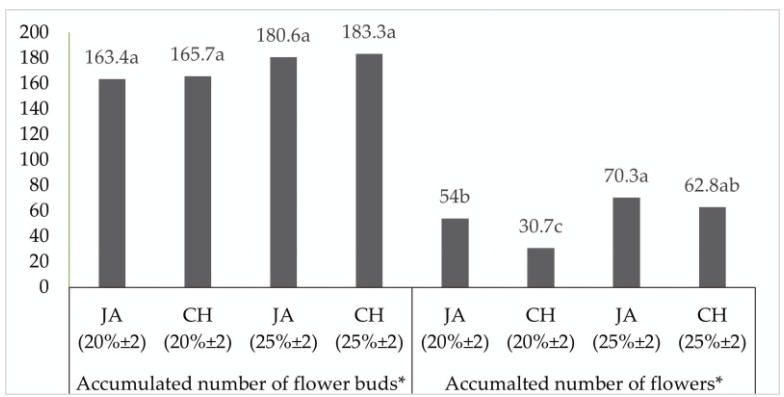

**Figure 9.** Effect of different soil water contents in two chili pepper morphotypes (*Capsicum annuum* L.) on accumulated number of flower buds per plant, and accumulated number of flowers per plant. * It is the sum of 10 evaluation dates with each 10 days apart. Tukey test ($p \leq 0.05$). Numbers with the same letter on bars in each variable are statistically equal. JA = Jalapeño; CH = Chilaca.

Regarding productivity variables, in terms of the accumulation of fruit yield in 10 harvest dates, the Jalapeño chili pepper produced the highest number of fruits per plant under OSMC with a value of 163.1, while he worst was the Chilaca chili pepper under SSMC with 90.9 fruits per plant, which is equivalent to a 44.26% decrease in the production of the latter. However, in terms of fruit weight per plant, the results are the contrary, where the treatment with the best response was for Chilaca in OSMC with a yield of 1191.6 g/plant, while the Jalapeño chili pepper reached 947.3 g/plant. In contrast, in SSMC, the Chilaca chili pepper had the lower yield with 567.5 g/plant, which is equivalent to a 52.37% decrease going from OSMC to SSMC. The higher yield of the Chilaca chili pepper was due to greater fruit size and weight compared to the Jalapeño chili pepper. Even so, the Jalapeño chili outperformed the Chilaca chili under SSMC conditions with a yield of 718.8 g/plant. Consequently, the yield ($kg/m^2$) was in the same proportion to the weight of the fruit yield/plant (Table 1).

**Table 1.** Effect to watered regimes in two chili pepper types (*Capsicum annuum* L.) on some productivity indicators (n = 16).

| Treatments: Chili Morphotypes—SMC | Number of Fruits per Plant | Weight of Harvested Fruits per Plant (g) | Chili Pepper Production * ($kg/m^2$) |
|---|---|---|---|
| JA—20% ± 2 | 126.8 [b] | 718.8 [bc] | 2.99 [bc] |
| CH—20% ± 2 | 90.9 [c] | 567.5 [c] | 2.36 [c] |
| JA—25% ± 2 | 163.1 [a] | 947.3 [b] | 3.94 [b] |
| CH—25% ± 2 | 107.1 [b] | 1191.6 [a] | 4.95 [a] |

Tukey test ($p \leq 0.05$). Numbers with the same letter in each column are statistically equal. SMC = Soil moisture content; JA = Jalapeño; CH = Chilaca. * According to 4.16 chili plants/$m^2$.

## 4. Discussion

The low rainfall that occurred during the experimental period did not influence the soil moisture contents since the irrigation was stopped when the precipitation occurred

in July. If the potential evaporation rate in the study area is 10 times greater than the rainfall [39], then the soil moisture content by a 54 mm rainfall is not maintained for more than five days in order to restore the soil moisture contents. This time period does not alter the growth and physiological indicators of the plant in the different treatments. After this time period, the irrigation was reinitiated according to the lower level soil moisture contents of 18 and 23% in OSMC and SSMC, respectively, reach higher levels of up to 22 and 27%, respectively. In the other months, the rainfall was minimum. The chili crop cycle is approximately 80 to 90 days after transplanting to the start of the fruiting stage, of which this time is from June to August, where the temperature varied from 27.6 to 25.9 °C, respectively, with a maximum temperature of 27.6 °C in June.

Findings about the response of the chili pepper morphotypes evaluated in this study in different soil moisture contents are congruent to those reported by diverse studies. Different water deficit responses at physiological, biochemical, cellular, and molecular levels are involved in plants tolerant to water deficits [46]. These processes include improvement in the root system, leaf structure, osmotic adjustment, relative water content, and stomata regulation of the gasses' flow in terms of photosynthesis, transpiration, and stomatal conductance rates [22,47]. RWC is an important physiological variable to know the tissue hydric status, and it is directly linked to plant water potential under different environments [48]. The last two measurements of the relative water content were coincident with the two measurements of the physiological indicators (30 and 75 DAT), which makes fit the relationship of these plant physiological processes through that time. The chili pepper morphotype Chilaca showed lower water retention under SSMC, and that could be linked to a higher water deficit tolerance [49], since different mechanisms are involved for the plants to tolerate dry environments [50].

In addition, different genes as transcription factors and signal transduction pathways are induced by water deficits. This is due to the capability of plants to activate genes as a result of diverse, physiological and biochemical attributes, thereby allowing plants to tolerate drought stress [51]. Soltys-Kalina et al. 2016 [52] found a difference in response to RWC in several potato genetic materials, which was the basis for a selection program for breeding.

Regarding the gas flow rates of photosynthesis and transpiration, the responses varied according to the soil moisture content and the chili pepper morphotypes used. Rosales et al. [53] reported that these physiological indicators can be induced for the accumulation of abscisic acid in plant tissues as a mechanism to tolerate water stress, but other mechanisms could be involved. Photosynthesis is the main basic activity to provide photo assimilates for biomass production fruiting and yield [54,55], and it is very sensitive to the environment and crop management. Particularly, environmental stress through temperature dynamics, light intensity, and low rainfall severely affects the plant physiological, biochemical, and molecular attributes, with an adverse impact on photosynthetic activity [50]. In this case, the Jalapeño chili pepper showed better stability of the photosynthesis rate to change from favorable to unfavorable water conditions, indicating that this morphotype is capable of maintaining high photosynthesis in both soil moisture contents evaluated in this study.

Transpiration and photosynthesis processes are correlated but they are differentially affected, even though both processes are affected by lower conductance due to stomatal closure under water deficits [56,57]. In this study, the Chilaca chili showed the highest stomatal sensitivity with a decreasing rate of photosynthesis at 75 DAT, which is possibly related to the start of the fruiting stage of the chili plant, and consequently, to the foliar senescence. Photosynthesis, as well as other physiological attributes such as stomatal conductance and transpiration, have been considered the most sensitive parameters to the environment, genotype, and its interaction [58]. Usually, leaf–gas interchange tends to decrease when plants are subjected to water deficits or another type of abiotic stress to improve the use of water inside the tissues, avoiding dehydration [24]. The ratio between photosynthesis and transpiration is named water use efficiency (WUE), and one plant is more efficient when the WUE is high, meaning that they consume more $CO_2$ with a lower

loss of water transpiring [59]. In this sense, the Chilaca chili pepper could have a better response to water deficits due its high sensitivity to stomatal closure; unfortunately, it was also negatively affected in the final yield under water deficit conditions.

About the productivity and yield variables, in this study, the high decrease in the flower buds occurred for an abortion of this flowering organs, which is natural in these types of crops, which could be to attenuate by avoiding water deficits or other limiting factors such as extreme maximum temperatures [60,61]. Most plants are highly sensitive to biotic and abiotic stress during the flowering stage. In accordance with this study, Sun et al. [62] and Moriyah and Irish [63] reported that water deficits cause flower buds' abortion in different crops. This type of response is common when soil moisture is low, with possible negative effects in productivity [64]. Drought stress reduces the yield by affecting the key plant metabolic pathways [51]. Water deficits, being an inevitable negative factor that exist in various environments, thereby hamper the plant biomass production and its quality [50].

According to the results in chili peppers' yield, if water availability is not limited, both chili pepper morphotypes produce adequately, of which Chilaca yielded 4.95 kg/m$^2$, followed by the Jalapeño chili with 3.94 kg/m$^2$. Under restricted water availability, Jalapeño could be the best option, since it is capable of maintaining its productivity when going from OSMC to SSMC, with a production of 2.99 kg/m$^2$. The Jalapeño chili reduced its yield by 24.1%, while Chilaca reduced its yield by 52.32%, going from an optimum to suboptimum soil moisture content. Similar results have been reported by Quintal et al. [33] for chili productivity under different soil moisture contents, where they found that 60% of the available water (AW) produced 55% more leaf area, 44% more total biomass, and 84% more fruit yield than with 20% AW.

This diversity of morphological, physiological, and productivity responses of a genetic type when going from a favorable to unfavorable hydric content is a property of most organisms, as they have different mechanisms for adapting to adverse conditions [24,30,65,66]. Some authors have reported that some crops have a quantitative benefit from production under favorable conditions, while under unfavorable conditions, such as water deficits and salinity in water and soil, the quality of production improves, but with a slight decrease in productivity [45,67]. Finally, since the results of physiological and productive indicators were obtained at an experimental area, these require to be validated in productive fields at an extensive level.

## 5. Conclusions

The Chilaca chili pepper was less stable in terms of relative water content, photosynthesis, stomatal conductance, and transpiration going from an optimum soil moisture content to a suboptimum soil moisture content, but with potential to support water deficits. The Jalapeño chili pepper showed more stability in the relative water content and recorded a greater photosynthetic rate and relatively low stomatal conductance and transpiration rates, producing a higher number of flowers per plant and number of fruits per plant, with a yield of 3.94 and 2.99 kg/m$^2$ going from an optimum to suboptimum soil moisture content, respectively. However, under the optimum soil moisture content, the Chilaca chili pepper produced a higher yield (4.95 kg/m$^2$) due to the greater fruit length and weight. Thus, the Jalapeño chili pepper could be an option when water is restricted, while the Chilaca chili pepper could be a better option for areas where irrigation water is not restricted.

**Author Contributions:** Conceptualization, A.P.-S., R.T.-C. and J.R.M.-F.; methodology, A.P.-S., R.T.-C. and I.G.-A.; software, A.P.-S. and R.T.-C.; validation, A.P.-S. and I.G.-A.; formal analysis, A.P.-S., J.R.M.-F. and R.T.-C.; investigation, A.P.-S., J.R.M.-F., R.T.-C. and I.G.-A.; resources, A.P.-S. and I.G.-A.; data curation, A.P.-S. and R.T.-C.; writing—original draft preparation, A.P.-S., J.R.M.-F. and R.T.-C.; writing—review and editing, A.P.-S., J.R.M.-F. and R.T.-C.; project administration, A.P.-S. and R.T.-C.; funding acquisition, A.P.-S., J.R.M.-F. and R.T.-C. All authors have read and agreed to the published version of the manuscript.

**Funding:** This research was supported by Dirección General de Investigación y Posgrado—Universidad Autónoma Chapingo through of the project ID: 22023-C-60.

**Institutional Review Board Statement:** Not applicable.

**Informed Consent Statement:** Not applicable.

**Data Availability Statement:** Data are contained within the article.

**Acknowledgments:** The authors thank José Antonio Miranda Rojas and Ramón Reyes Urias for Technical support. In addition, we give thanks to the Water Soil Plant and Atmosphere Laboratory of the National Institute of Forestry, Agricultural and Livestock Research for the soil analysis.

**Conflicts of Interest:** The authors declare no conflicts of interest.

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
