# Peer review of "Physiological and Productivity Responses in Two Chili Pepper Morphotypes (Capsicum annuum L.) under Different Soil Moisture Contents"

_horticulturae, doi:10.3390/horticulturae10010092_

Round 1

Reviewer 1 Report

Comments and Suggestions for Authors

In the Abstract part:

Must define precisely the area of the pots (small and large is not enough)!

After 25%±2, and 20%±2 must write the meaning, i.e. that is the soil moisture content!

OWR, SOWR are not defined: what are these abbreviation mean?

In the Introduction part:

Use United Nations instead of Union Nations.

Define differences between the used morphotypes!

In the Results part:

In Fig. 3. Correct the month July!

Use the rate of photosynthesis instead of rate photosynthesis!

What does DAT mean in Fig. 5. Days after transplanting? That must precisely define!

Explain in this part or in the part “Discussion” why decreased the rate of photosynthesis and the transpiration at 75 DAT in case of chilaca morhotype? May be that senescence started?

In row 256: Use Transpiration behaved similarly…

In rows 258-259: Data are not fit to the numbers in Fig. 6. Change to the correct values!

In row 283: Use flower bud!

In row 361-362: …chilaca showed higher stomatal sensitivity.. There were no investigation on stomatal conductance, so where is this statement originating from?

Generally:

How long was it possible to detect the effect of the high rainfall in July, and how did it affect the results of the experiment?

To what extent was the determination of the relative water content of the leaves consistent with the timing of the other measurements (photosynthesis, transpiration)?

Is it sufficient to examine data from 1 growing season to compare drought tolerance, taking into account potential temperature variations?

Reviewer 2 Report

Comments and Suggestions for Authors

1.There is a lack of previous research on the impact of soil moisture on chili peppers in the research background. Meanwhile, the two chili varieties in this experiment were not seen in the background, and these two chili varieties have the largest planting area in Mexico? Or is the water demand lower? Why choose these two varieties for research? This makes readers confused about the selection of experimental varieties.

2. The manuscript lacks a map of the experimental points. It is recommended to add it to facilitate readers' understanding of the overview of the experimental points.

3.What are the physical and chemical properties of the experimental soil? For example, soil type, soil fertility, bulk density and other indicators, it is recommended that the author add relevant information.

4.Is there a reference basis for the moisture control interval in experimental design and treatment? Why is there only two moisture gradients designed in the experiment?

5.Method for measuring moisture characteristic curve? Why is it a quadratic curve? Usually in exponential form.

6.The manuscript lacks a method for detecting the photosynthetic index of chili peppers, with detailed calculation methods. It is recommended that the author supplement the method and formula for detecting the index. For non original methods, please add references.

7.What is the irrigation system during the experiment? Please ask the author to add an irrigation system for reference by other researchers.

8.    Why choose these two periods to measure the photosynthetic indicators? Is there any reference basis?

9.    The manuscript only includes precipitation data during the experimental period. It is suggested to supplement temperature data for the reference of other scholars.

10.    The conclusion is not concise enough. Please simplify the conclusion.

Comments on the Quality of English Language

1.There is a lack of previous research on the impact of soil moisture on chili peppers in the research background. Meanwhile, the two chili varieties in this experiment were not seen in the background, and these two chili varieties have the largest planting area in Mexico? Or is the water demand lower? Why choose these two varieties for research? This makes readers confused about the selection of experimental varieties.

2. The manuscript lacks a map of the experimental points. It is recommended to add it to facilitate readers' understanding of the overview of the experimental points.

3.What are the physical and chemical properties of the experimental soil? For example, soil type, soil fertility, bulk density and other indicators, it is recommended that the author add relevant information.

4.Is there a reference basis for the moisture control interval in experimental design and treatment? Why is there only two moisture gradients designed in the experiment?

5.Method for measuring moisture characteristic curve? Why is it a quadratic curve? Usually in exponential form.

6.The manuscript lacks a method for detecting the photosynthetic index of chili peppers, with detailed calculation methods. It is recommended that the author supplement the method and formula for detecting the index. For non original methods, please add references.

7.What is the irrigation system during the experiment? Please ask the author to add an irrigation system for reference by other researchers.

8.    Why choose these two periods to measure the photosynthetic indicators? Is there any reference basis?

9.    The manuscript only includes precipitation data during the experimental period. It is suggested to supplement temperature data for the reference of other scholars.

10.    The conclusion is not concise enough. Please simplify the conclusion.

Reviewer 3 Report

Comments and Suggestions for Authors

In this study, the effects of different soil moisture content on the physiology and productivity of two kinds of chili pepper morphotypes (Capsicum annum L.) were analyzed through randomized trials. I suggest making corrections from the following aspects:

1. The morphological difference between the two kinds of pepper and the mechanism of the response of morphology to soil water content need to be further explained.

2. What is the basis for setting the soil moisture content in the design of the experiment, and the proposed explanation and explanation.

3. Whether the number of variables of soil water content is too small to fully include the response law of pepper physiology and productivity to soil water content, it is suggested to explain the limitations of this study

4. The discussion section suggests greater logic and generality.

Round 2

Reviewer 2 Report

Comments and Suggestions for Authors

1.How did the author monitor soil moisture during the experiment? How do you know when to irrigate? What are the irrigation frequency and water quantity? Suggest the author to supplement the manuscript in its entirety.

2.Is there a reference basis for the two irrigation levels in the experimental design?What is the basis for determining OSMC and SSMC?

3.Figure 1 suggests the author to add latitude and longitude to facilitate readers to better understand the content of the image and the location of the experimental points.

4.The pressure in Figure 3 is only -2 Mpa, resulting in incomplete soil moisture characteristic curve and low fitting degree (R2). It is recommended that the author supplement experiments to improve the moisture characteristic curve.

5.What does "~25 cm" mean on line 184? Suggest checking for text errors throughout the entire text.

6.Is there a lack of research progress on water control in pepper irrigation in the introduction section? Suggest the author to add.

Round 3

Reviewer 2 Report

Comments and Suggestions for Authors

The author made good revisions according to the suggestions
